# Atoh8 acts as a regulator of chondrocyte proliferation and differentiation in endochondral bones

Nadine Schroeder[1], Manuela Wuelling[1], Daniel Hoffmann[2], Beate Brand-Saberi[3], Andrea Vortkamp[1]*

**1** Center for Medical Biotechnology, Department of Developmental Biology, University of Duisburg-Essen, Essen, Germany, **2** Center for Medical Biotechnology, Bioinformatics and Computational Biophysics, University of Duisburg-Essen, Essen, Germany, **3** Department of Anatomy and Molecular Embryology, Ruhr-University Bochum, Bochum, Germany

* Andrea.Vortkamp@uni-due.de

## Abstract

Atonal homolog 8 (Atoh8) is a transcription factor of the basic helix-loop-helix (bHLH) protein family, which is expressed in the cartilaginous elements of endochondral bones. To analyze its function during chondrogenesis we deleted Atoh8 in mice using a chondrocyte- (Atoh8$^{flox/flox}$; Col2a1-Cre) and a germline- (Atoh8$^{flox/flox}$;Prx1-Cre$^{female}$) specific Cre allele. In both strains, Atoh8 deletion leads to a reduced skeletal size of the axial and appendicular bones, but the stages of phenotypic manifestations differ. While we observed obviously shortened bones in Atoh8$^{flox/flox}$;Col2a1-Cre mice only postnatally, the bones of Atoh8$^{flox/flox}$;Prx1-Cre$^{female}$ mice are characterized by a reduced bone length already at prenatal stages. Detailed histological and molecular investigations revealed reduced zones of proliferating and hypertrophic chondrocytes. In addition, Atoh8 deletion identified Atoh8 as a positive regulator of chondrocyte proliferation. As increased Atoh8 expression is found in the region of prehypertrophic chondrocytes where the expression of Ihh, a main regulator of chondrocyte proliferation and differentiation, is induced, we investigated a potential interaction of Atoh8 function and Ihh signaling. By activating Ihh signaling with Purmorphamine we demonstrate that Atoh8 regulates chondrocyte proliferation in parallel or downstream of Ihh signaling while it acts on the onset of hypertrophy upstream of Ihh likely by modulating Ihh expression levels.

## Introduction

During endochondral ossification, bones are formed by a multistep process, which includes the formation of a cartilage template of the later skeletal element and its subsequent replacement by bone. The cartilage anlagen originate in mesenchymal cells, which condense and differentiate into chondrocytes. These chondrocytes proliferate and express the extracellular matrix protein Collagen type 2 (Col2) [1]. Two subtypes of proliferating chondrocytes can be distinguished: round, slow proliferating cells at the end of the cartilage elements (round/resting chondrocytes) and flat, highly proliferating cells organized in columns towards the

**Funding:** This work was supported by the Mercator Research Center Ruhr (grant no. Pr2012-0058) to AV and BBs. The funder had no role in study design, data collection and analysis, decision to publish or preparation of the manuscript.

**Competing interests:** The authors have declared that no competing interests exist.

hypertrophic region (columnar chondrocytes). When the cartilage anlagen reach a critical size, proliferating chondrocytes in their center exit the cell cycle and differentiate into Indian hedgehog (Ihh) producing, prehypertrophic [2, 3] and, subsequently, Collagen type 10 (Col10) expressing, hypertrophic chondrocytes [4]. Eventually, blood vessels invade the zone of hypertrophic chondrocytes and the hypertrophic cells are replaced by bone and bone marrow. Postnatally, secondary ossification centers (SOC) are formed at the ends of the endochondral long bones. Between the two regions of ossification, parts of the embryonic cartilage, the so-called growth plates, remain to organize longitudinal growth after birth [5]. As longitudinal bone growth depends on the balance between chondrocyte proliferation and hypertrophic differentiation, both processes are tightly regulated. Although many regulators have been identified, clarifying the detailed molecular mechanisms is still in progress.

Atonal homolog 8 (Atoh8, also known as Math6 in mouse) is a transcription factor of the basic helix-loop-helix (bHLH) protein family [6]. The helix-loop-helix region of these proteins mediates the interaction with other bHLH proteins, while their basic region binds to a specific DNA sequence, the E-box element [7].

In contrast to other atonal-related proteins, which show a tissue restricted pattern of expression, Atoh8 is broadly expressed in many organs including the brain, kidney and skeletal muscles and regulates proliferation and differentiation of distinct cell types. For instance, overexpression of Atoh8 in retinal explant cultures promotes the differentiation of neuronal progenitors towards the neuronal versus the glial cell lineage [6]. Consistent with this, Atoh8 is necessary for the differentiation of the hypaxial muscles of the trunk in chicken embryos, which underpins its importance for the proliferation-to-differentiation transition in embryonic development [8]. In mice, loss of Atoh8 expression leads to podocyte dedifferentiation during kidney development [9]. In addition, Atoh8 has been described as an activator of proliferation during muscle regeneration [10] and in colorectal cancer [11].

In the developing skeleton, Atoh8 has been shown to be expressed in avian growth plates [12] and in mouse limb buds [13, 14], but its molecular role has not been investigated.

In this study, we show that Atoh8 deficiency results in reduced zones of proliferating and hypertrophic chondrocytes and, consequently, in a decreased length of endochondral bones. In addition, we observed a reduced proliferation rate due to Atoh8 depletion suggesting Atoh8 positively regulates chondrocyte proliferation. Moreover, the altered domain of proliferating chondrocytes indicates an additional function as a regulator of hypertrophy. In limb explant cultures, we found Atoh8 to regulate the onset of chondrocyte differentiation upstream of Ihh, while it acts independent or downstream of Ihh in the regulation of proliferation.

## Materials and methods

### Transgenic mice and genotyping

All animal experiments were performed according to the institutional guidelines of the University Duisburg-Essen, specifically approved by the animal welfare officer of the University Duisburg-Essen. Mouse husbandry was approved by the city of Essen (Az: 32-2-11-80-71/348) in accordance with § 11-1a of the "Tierschutzgesetz". Work with transgenic animals and BrdU injection was approved by the "Bezirksregierung Duesseldorf" (Az: 53.02.01-D-1.55/12, Anlagen-Nr. 1464 (transgenic) and 84–02.05.20.12.157, Anlage-Nr. B1294/12 (BrdU)) in accordance with § 8 Abs. 4 Satz 2 GenTG of the "Gentechnikgesetz". *Col2a1-Cre* [15] or *Prx1-Cre* [16] mice were interbred with *Atoh8$^{flox/flox}$* mice [17] and the resulting *Atoh8$^{flox/+}$;Col2a1-Cre* or *Atoh8$^{flox/+}$; Prx1-Cre$^{female}$* mice were backcrossed to *Atoh8$^{flox/flox}$* mice to generate *Atoh8$^{flox/flox}$;Col2a1-Cre* or *Atoh8$^{flox/flox}$;Prx1-Cre$^{female}$* mutants. As controls, corresponding littermates without the *Col2a1-Cre* or *Prx1-Cre* allele were used. In addition, heterozygous *Atoh8$^{flox/+}$;Col2a1-Cre* or

*Atoh8^{flox/+};Prx1-Cre^{female}* littermates were analyzed as Cre-positive controls. All strains were maintained on a C57BL/6J genetic background. Genotyping of tail DNA was performed using the following primers: *Atoh8*, forward (Atoh8-fwd): 5´-GGAAAGTTCCTAC TCGTCAATTTCA CACG-3´ and reverse (Atoh8-rev): 5´-CCAAGGGACCAGAAGACAAATA CTCGC-3´; *Col2a1-Cre* or *Prx1-Cre* transgene, forward: 5´-GAGTGATGAGGTTCGCAAGA-3´ and reverse: 5´-CTACACCAGAGACGG-3´; to control Cre mediated excision, forward (Ex-fwd): 5´-AGCCTCTTCTTTCTAGTAGGATTTCCAGTGG-3´ and reverse (Ex-rev): 5´-CTCTG TCCTTCTCAGTCAACAAGATGATGTC-3´. Wild-type mice (NMRI) were obtained from Charles River Laboratories.

## Histology

For histological analysis, forelimbs of E12.5-E18.5 embryos and P7-P14 mice were fixed overnight in 4% paraformaldehyde at 4˚C. Forelimbs of P7-P14 mice were decalcified in 25% EDTA for at least one week at 37˚C. After dehydration and embedding in paraffin, serial sections of 4–5 μm (prenatal stages) or 7 μm (postnatal stages) were stained with Safranin-Weigert as described previously [18] or used for *in situ* hybridization or BrdU labeling.

All length measurements were carried out on sections covering the entire skeletal element.

## *In situ* hybridization

For fluorescence *in situ* hybridization, RNA antisense probes were labeled with Digoxigenin-11-UTP (Roche). Hybridization was performed as described previously [19]. DNA was counterstained with 5 mg/ml DAPI (Roth).

For whole-mount *in situ* hybridization, RNA antisense and sense probes were labeled with Digoxigenin-11-UTP (Roche). Embryo preparation and hybridization was performed as described previously [20] using α-Digoxigenin-AP (Roche) and NBT-BCIP solution (Sigma) for detection [21].

The following probes were used for *in situ* hybridization: rat *Col2a1* [22], mouse *Ihh* [23], mouse *Col10a1* [24] and mouse *Atoh8* using a 770 bp fragment corresponding to nucleotides 445–1214 of mouse Atoh8 mRNA subcloned into pDrive.

For measurements, 2 – 6 sections were analyzed per animal.

## Skeletal preparation

Six month old, male mice were skinned, eviscerated and fixed in 100% ethanol overnight at 4˚C. Alcian blue and Alizarin red staining and subsequent clearing in 1% potassium hydroxide were carried out as described previously. Samples were stored in 100% glycerol [25].

## BrdU labeling

To analyze chondrocyte proliferation pregnant (14.5 days post fertilization) and P7 mice received an intraperitoneal injection of 16 μl/kg and 20 μl/kg body weight BrdU (10 mM), respectively, 2 h before being sacrificed as described previously [26]. Limb explants were treated for 2 h with 0.1 mM BrdU. Sections of forelimbs or limb explants were stained with a rat α-BrdU (Abcam) antibody and detected with an Alexa 568 goat α-rat (Life Technologies) secondary antibody. DNA was counterstained with 5 mg/ml DAPI (Roth). BrdU labelled and total cell numbers were scored in defined regions of 3–9 sections per animal using the Metamorph imaging software (Visitron Systems) or a combination of Fiji and CellProfiler.

## Limb explant cultures

For limb explant cultures, forelimbs were dissected from E16.5 embryos and skin and muscle tissue were removed. Limb explants were cultured for 48 h (proliferation analysis) or 96 h (*in situ* hybridization) as described previously [27]. One forelimb of each embryo was treated with 4 µM Purmorphamine (Abcam) while the second forelimb served as a control and was treated with an equivalent amount of DMSO (AppliChem). After culture, limb explants were fixed and embedded.

## Micromass cultures and Alcian blue staining

Primary chondrocytes were isolated from limb buds of E12.5 embryos by digestion with 1 U/ml Neutral Protease NB (Serva) for 15 min and with 0.3 U/ml collagenase NB 4 (Serva)/0.05% trypsin-EDTA (Life Technologies) for 30 min at 37˚C. Single cell suspensions were obtained by passing through a 40-µm cell strainer. $2 \cdot 10^7$ cells were cultured in high-density as described previously [28]. For differentiation into hypertrophic chondrocytes, micromass cultures were cultured for 28 days. Afterwards, micromass cultures were fixed and stained with Alcian blue [28].

## Cell culture

ATDC5 cells [29] were cultured in DMEM (Thermo Fisher Scientific) with 5% fetal calf serum (PAN-Biotech) and 1% Penicillin/Streptomycin (Thermo Fisher Scientific). The human Atoh8 expression plasmid [30] was transfected by electropulsing using the Cell Line Nucleofector System (Lonza) following standard procedures provided by the manufacturers.

## Quantitative reverse transcription PCR (qRT-PCR)

RNA was extracted from micromass cultures, embryos or forelimb skeletal elements using the NucleoSpin RNA kit (Macherey-Nagel). cDNA was synthesized with the Maxima First Strand cDNA Synthesis Kit for RT-qPCR (Thermo Scientific). qRT-PCR was performed with a 1:2 dilution of cDNA by using the StepOnePlus Real-Time PCR-System (Thermo Fisher Scientific) or the CFX384 Touch™ Real-Time PCR detection system (Bio-Rad) with the my-Budget 5x EvaGreen (R) QPCR-Mix II (BioBudget). All mRNA quantification data were normalized to Beta-2 microglobulin (B2M). Relative quantity of mRNA was calculated by the ΔCt method. The following primer pairs were used: for Atoh8, forward: 5´-CTTCGAGGCGCTGAGGAAG–3´ and reverse: 5´-GCAGGTCACTCCTTCCGTTT–3´; for Col2, forward: 5´-CGAGTGGAA GAGCGGAGACT–3´ and reverse: 5´-AACTTTCATGGCGTCCAAGGT–3´; for Ihh, forward: 5´-TCAGACCGTGACCGAAATAA–3´ and reverse: 5´-ACACGCTCCCCGTTCTCTA–3´; for B2M, forward: 5´-TCGGCGCTTCAGTCGCGGTCG–3´ and reverse: 5´-TCCCATTCTCCGGT GGGTGGCGT–3´. Additional primer pairs used for S4 Fig are listed in S2 Table.

## Imaging

Bright and dark field pictures were taken on a Zeiss Axioplan 2 microscope with a SPOT 14.2 camera and SPOT advanced software (Diagnostic Instruments), radioactive *in situ* hybridization signals were visualized using the illuminator Intralux 5000–1 (Volpi). Fluorescence pictures were taken on a Zeiss Axiovert 200 microscope with a Spot 23.0 camera (Diagnostic Instruments) and Metamorph imaging software (Visitron Imaging Systems) or a Zeiss Axio Observer 7 microscope with an AxioCam 506 mono CCD camera (Zeiss) and Zen 2.3 software (Zeiss).

## Statistical analysis

Probabilities $p_-$ and $p_+$ (effect direction probabilities) that Atoh8 deletion is associated with a negative or positive effect, respectively, on the measured quantities were estimated by Bayesian inference with Generalized Linear Models (GLMs). Specifically, if the measured quantities were continuous and positive (distances, or, in qPCR measurements, relative expressions), we modeled the measurements as generated by a linear model linked to the measurement scale by a logarithm, and with a normal noise distribution; if the measured quantities were counts (BrdU experiments), measurements were considered as generated by a linear model linked by a logit-function to a binomial probability parameter, and finally a binomial distribution producing the observed data. In all linear parts of the GLMs, effects coming from batches, litters, or animals were treated as separate intercepts in the linear model. Priors for all fitted parameters were chosen much broader than the distribution of the actual data so that the posteriors are mainly determined by the data. Models were fitted with the default sampler of the Stan software, version 2.17 [31], typically with three Markov chains, each of several thousand iterations. Convergence of Markov chains was monitored with the Gelman-Rubin diagnostic parameter , which was lower than 1.02 in all cases, and the effective sample size, typically in the hundreds or thousands. Posterior predictive checks were carried out to test consistency of all fitted models with the data of the respective experiments. Finally, $p_-$ and $p_+$ values were computed by sampling from the fitted posterior the strength of the effect of interest (e.g. the effect of the genotype) on the measured quantity. All $p_-$ and $p_+$ values are listed in S1 Table. The interpretation of effect direction probabilities $p_-$ and $p_+$ is simple: e.g. if we are interested in the effect of Atoh8 on Col2 expression (Fig 1E), the value $p_-$ = 0.984 at day 7 means that a negative effect of the Atoh8-- knockout on Col2 expression has a probability of 0.984; accordingly, the probability of a positive effect is $p_+$ = 1-$p_-$ = 0.016. All effect direction probabilities are given to 3 digits. Effect direction probabilities $p_- \geq 0.950$ and $p_+ \leq 0.050$ or vice versa are regarded to be significant. To analyze if the effects of the Purmorphamine treatment varied between the genotypes, two GLMs were developed. In one model genotype and treatment effects were added with an interaction and in the other model without an interaction. We then compared the two models for their ability to generalize to unseen data by an approximate leave-one-out cross-validation [32]. Data in bar plots are presented as mean ± standard deviation. Standard deviations are shown as error bars in all bar plots. n always corresponds to biological replicates.

## Results

### Atoh8 is expressed in proliferating and hypertrophic chondrocytes

To receive first insight into the role of Atoh8 in skeletal development, we investigated its detailed expression pattern in mouse skeletal tissues. Whole-mount *in situ* hybridization of E12.5 embryos confirmed the expression in early limb buds, particularly in the developing cartilage anlagen (Fig 1A–1C) [13, 14]. In order to determine the specific expression of Atoh8 in chondrocytes, we compared its expression to that of Col2 and Col10, markers of proliferating and hypertrophic chondrocytes, respectively. *In situ* hybridization on parallel sections of E14.5 wild-type limbs revealed that Atoh8 is expressed in both, proliferating and hypertrophic chondrocytes (Fig 1D and 1E) with an increased level of expression in prehypertrophic chondrocytes (Fig 1D, arrows). Furthermore, we analyzed Atoh8 expression during the differentiation of primary chondrocytes in micromass cultures by qRT-PCR. High Atoh8 expression was detected at day 7 of culture, while it continued to be expressed at lower levels until at least day 28 when hypertrophic differentiation has taken place as demonstrated by the decrease in Col2 expression (Fig 1F and 1G).

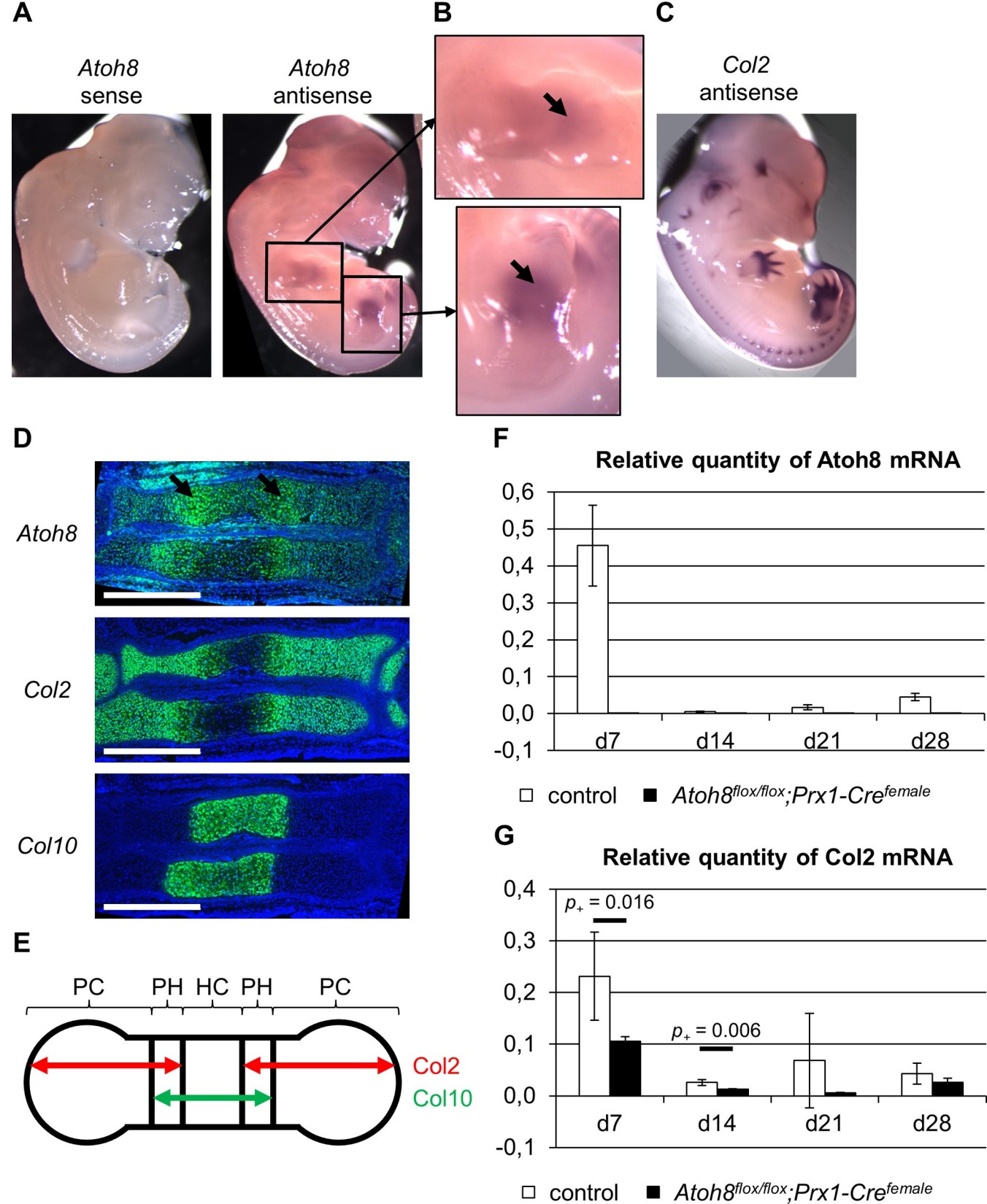

**Fig 1. Atoh8 expression profile.** (A) Whole-mount *in situ* hybridization of E12.5 wild-type embryos with an *Atoh8* sense and antisense probe reveals Atoh8 expression in mouse limb buds. (B) A magnified view of the limb buds framed in A shows Atoh8 expression in the cartilage anlagen (black arrows) overlapping with the expression of Col2 (C). (D) Parallel sections of an E14.5 wild-type forelimb hybridized with *Atoh8*, *Col2* and *Col10* antisense probes. Atoh8 is expressed in proliferating and hypertrophic chondrocytes with an increased expression in prehypertrophic chondrocytes (black arrows). (E) Schematic representation of a developing long bone at E14.5 to visualize Col2 (red left-right arrows) and Col10 (green left-right arrow) expression domains and the zones of different chondrocyte populations. PC = proliferating chondrocytes, PH = prehypertrophic chondrocytes, HC = hypertrophic chondrocytes. (F, G) Relative Atoh8 (F) and Col2 (G) mRNA expression of chondrocytes cultured in micromass culture for up to 28 days in differentiation medium. Primary chondrocytes were isolated from E12.5 *Atoh8$^{flox/flox}$;Prx1-Cre$^{female}$* embryos and control littermates. n = 4 control and 3 *Atoh8$^{flox/flox}$;Prx1-Cre$^{female}$* mice from 1 litter; Bayesian analysis. Scale bar (white lines) in D: 500 μm.

In contrast to Atoh8, no other *Atoh* gene is expressed in chondrocytes, as no noteworthy mRNA expression is detectable via qRT-PCR in control or Atoh8-deficient developing skeletal elements (S4 Fig).

## Chondrocyte-specific deletion of Atoh8 results in reduced bone length due to disturbed chondrocyte proliferation and differentiation at postnatal stages

In order to determine the role of Atoh8 during endochondral ossification we analyzed mice carrying a chondrocyte-specific deletion of Atoh8. For this purpose, we used a conditional allele of Atoh8, in which Exon1 is flanked by *loxP* sites (S1 Fig) [17], and deleted Exon1 of *Atoh8* specifically in chondrocytes using a Col2a1-Cre driver [15]. Both, hetero- and homozygous mutants were born in Mendelian ratios and were viable and fertile. Besides a slightly reduced body size, *Atoh8$^{flox/flox}$;Col2a1-Cre* mutants did not show any obvious phenotypes.

To investigate the skeletal phenotype of the mutants in detail, we analyzed Safranin-Weigert stained forelimb sections of embryonic and juvenile mice and Alcian blue/Alizarin red stained adult skeletons. We could not detect distinct differences between mutant and control mice at prenatal stages (S3 Fig). However, from early postnatal stages (P7 and P14) until adulthood (6 month) the skeletal size was reduced in *Atoh8$^{flox/flox}$;Col2a1-Cre* mice (Fig 2). Measuring the length of the radius as a representative bone revealed a slight, but significant reduction in *Atoh8$^{flox/flox}$;Col2a1-Cre* mice at P7 (-13.5%, $p_+ < 0.001$), P14 (-14.7%, $p_+ < 0.001$) and 6 month (-7.3%, $p_+ = 0.003$) (Fig 2B and 2C). This reduction in bone length is more pronounced in mice with homozygous than with heterozygous Atoh8 deletion (S2A Fig).

To determine the molecular basis for the reduced bone length in *Atoh8$^{flox/flox}$;Col2a1-Cre* mice, we analyzed the size of the cartilage region including proliferating and hypertrophic chondrocytes on sections of P7 mice after Safranin-Weigert staining. Compared to control littermates, we found it to be shortened in *Atoh8$^{flox/flox}$;Col2a1-Cre* mice (Fig 3A top) indicating that the reduced bone length is due to impaired chondrocyte proliferation and/or differentiation. To investigate which step of the differentiation program is affected, we analyzed the expression of Col2 and Col10, markers for proliferating and hypertrophic chondrocytes, respectively, on parallel sections of P7 forelimbs by *in situ* hybridization. In order to determine if the onset of hypertrophy is disturbed, we measured the distance of the Col10 expression zone to the joint as well as the length of the Col2 expression zone, which demarcate the region of proliferating chondrocytes. Both zones were decreased in P7 *Atoh8$^{flox/flox}$;Col2a1-Cre* mutants ($p_+ = 0.009$ and $p_+ = 0.025$) (Fig 3B–3D) pointing to a premature onset of hypertrophic differentiation. Furthermore, the zone of Col10 expressing hypertrophic chondrocytes was shortened in P7 *Atoh8$^{flox/flox}$;Col2a1-Cre* mutants ($p_+ = 0.022$) (Fig 3B–3D) indicating that in addition the process of hypertrophic differentiation is accelerated. To investigate, whether a reduced proliferation rate also contributes to the reduced skeletal size, we analyzed cells in S-phase after bromodesoxyuridine (BrdU) incorporation. We found that the proliferation rate of

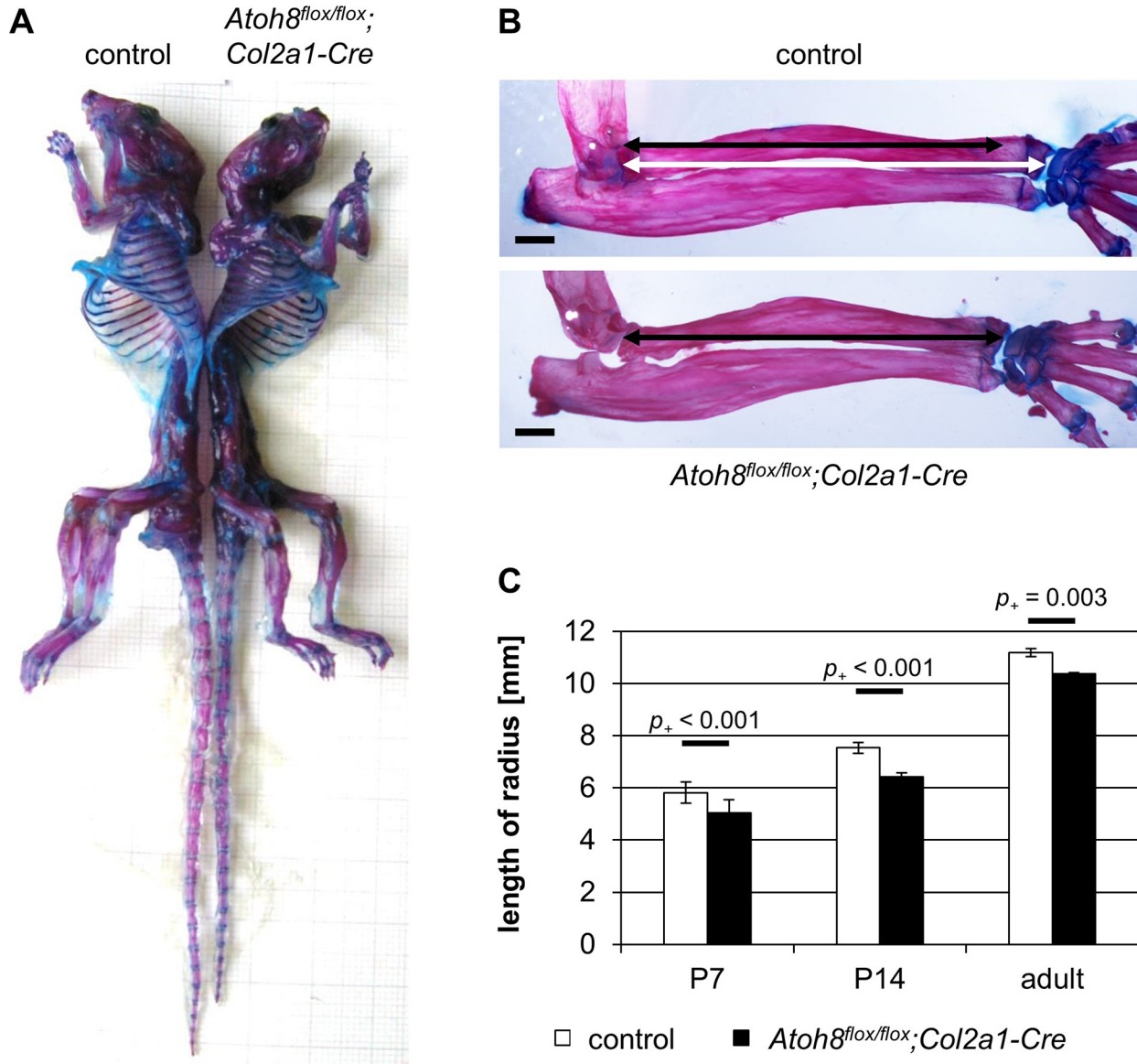

**Fig 2. Postnatal *Atoh8^flox/flox^;Col2a1-Cre* mice show a reduced skeletal size.** (A) Skeletal preparations of six month old mice show that *Atoh8^flox/flox^;Col2a1-Cre* mutant mice are smaller than their control littermates. (B, C) The radius of adult (6 month) and juvenile (P7, P14) *Atoh8^flox/flox^;Col2a1-Cre* mutants is significantly shorter than that of control littermates. (B) Measurements for C are visualized with white (control) and black (mutant) left-right arrows. n = 10 control and 8 *Atoh8^flox/flox^;Col2a1-Cre* mice from 5 litters (P7), n = 8 control and *Atoh8^flox/flox^;Col2a1-Cre* mice from 5 litters (P14), n = 3 control and *Atoh8^flox/flox^;Col2a1-Cre* mice from 2 litters (6 month); Bayesian analysis. Scale bar (black lines) in B: 1 mm.

round and columnar proliferating chondrocytes is reduced by 1.4 ($p_+ = 0.003$) and 5.4% ($p_+ = 0.004$), respectively, in *Atoh8^flox/flox^;Col2a1-Cre* mutants (Fig 3E and 3F).

Besides the reduced region of proliferating chondrocytes, Safranin-Weigert stained forelimb sections of *Atoh8^flox/flox^;Col2a1-Cre* mice showed a reduced zone of hypertrophic chondrocytes at the ends of the long bones, where the formation of the SOC is initiated, at P7 (Fig 3A top, arrows) and, consequently, a reduced SOC at P14 (Fig 3A bottom, arrows).

Taken together, our analyses identified Atoh8 as a regulator of chondrocyte proliferation and differentiation during postnatal bone growth.

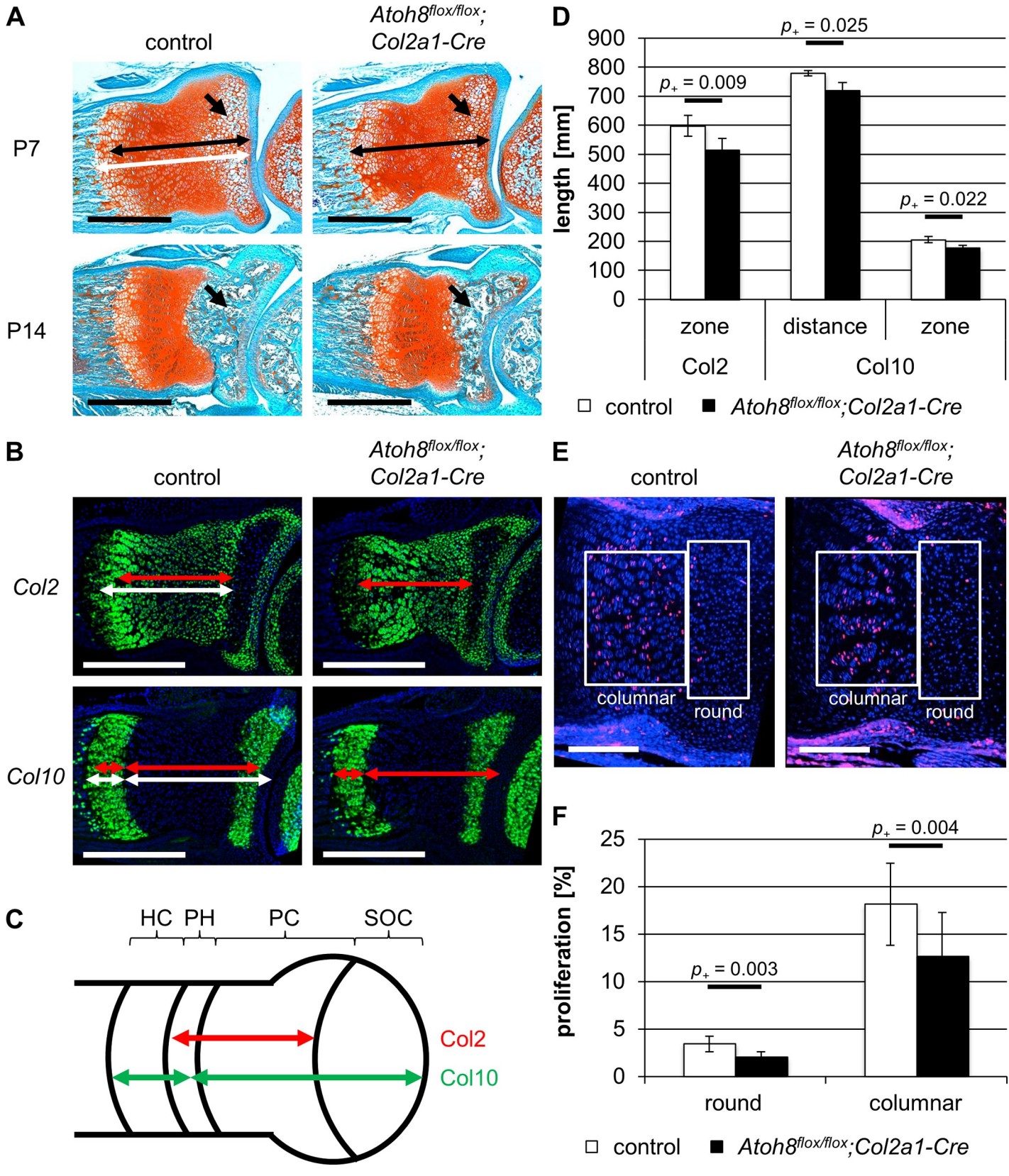

**Fig 3. *Atoh8$^{flox/flox}$;Col2a1-Cre* mice show reduced chondrocyte proliferation and accelerated hypertrophic differentiation at postnatal stages.** (A) Sections of P7 and P14 control and *Atoh8$^{flox/flox}$;Col2a1-Cre* forelimbs stained with Safranin-Weigert. The zone of chondrocytes in the radius is shortened in P7 *Atoh8$^{flox/flox}$;Col2a1-Cre* mutants as visualized by white (control) and black (mutant) left-right arrows. Furthermore, the formation of the SOC (black arrows) is delayed in *Atoh8$^{flox/flox}$; Col2a1-Cre* limbs. (B-D) *In situ* hybridization of parallel control and *Atoh8$^{flox/flox}$;Col2a1-Cre* forelimb sections with a *Col2* and *Col10* antisense probe revealed an accelerated onset and process of hypertrophic differentiation in P7 *Atoh8$^{flox/flox}$;Col2a1-Cre* mice. Measurements for D are visualized with white (control) and red (mutant) left-right arrows (B) and with red (Col2) and green (Col10) left-right arrows in the schematic representation of the developing long bone (C). (C) PC = proliferating chondrocytes, PH = prehypertrophic chondrocytes, HC = hypertrophic chondrocytes, SOC = secondary ossification center. (E, F) BrdU labeling of P7 control and *Atoh8$^{flox/flox}$;Col2a1-Cre* forelimb sections revealed a significantly reduced proliferation rate in round and columnar chondrocytes in *Atoh8$^{flox/flox}$;Col2a1-Cre* mice. Regions of round and columnar chondrocytes analyzed in F are framed with white boxes. (D) n = 4 control and 3 *Atoh8$^{flox/flox}$;Col2a1-Cre* mice from 2 litters; (F) n = 6 control and 4 *Atoh8$^{flox/flox}$;Col2a1-Cre* mice from 3 litters; (D, F) Bayesian analysis. Scale bar in A (black lines) and B (white lines): 500 μm and in E (white lines): 200 μm.

## Ubiquitous deletion of Atoh8 reduces chondrocyte proliferation and accelerates the onset of hypertrophic differentiation at prenatal stages

To decipher if ubiquitous deletion of Atoh8 results in a stronger phenotype than the chondrocyte-specific deletion, we used a female-transmitted *Prx1-Cre* allele, which leads to the expression of Cre recombinase in the germline [16]. Both, heterozygous and homozygous mutants were viable. Genotyping (S1 Fig) of tissue from the tail, in which Prx1 is not expressed, confirmed the ubiquitous recombination of the allele. Furthermore, qRT-PCR of micromass cultures, which besides chondrocytes contain a mixed population of mesenchymal cells, did not detect Atoh8 mRNA expression in cultures of *Atoh8$^{flox/flox}$;Prx1-Cre$^{female}$* mice (Fig 1F).

To detect differences in size of the embryonic skeleton, we measured the length of the radius on Safranin-Weigert stained sections of E16.5 (Fig 4A) and E14.5 (Fig 5A) embryos. We found that the radius of *Atoh8$^{flox/flox}$;Prx1-Cre$^{female}$* mice was significantly shorter than that of control and *Atoh8$^{flox/+}$;Prx1-Cre$^{female}$* mice at both embryonic stages with $p_+ = 0.005/0.027$ for E16.5 (Fig 4A and 4B and S2C Fig) and $p_+ < 0.001$ for E14.5 (Fig 5A and 5B and S2B Fig).

To investigate if Atoh8 also affects the differentiation of chondrocytes prenatally, we analyzed the regions of proliferating and hypertrophic chondrocytes after *in situ* hybridization with *Col2*, *Ihh* and *Col10* antisense probes (Fig 4C) in *Atoh8$^{flox/flox}$;Prx1-Cre$^{female}$* mice at E16.5. Measuring the size of the Col2 expression domain ($p_+ = 0.016$) and the distance of the Col10 ($p_+ < 0.001$) and Ihh ($p_+ = 0.020$) expression domains to the joint, we found a reduced region of proliferating chondrocytes at E16.5, while the region of hypertrophy demarcated by the Col10 expression domain was also reduced at this stage but not as clearly ($p_+ = 0.062$) (Fig 4E).

Morphological analysis of Safranin-Weigert stained sections of E14.5 control and *Atoh8$^{flox/flox}$;Prx1-Cre$^{female}$* forelimbs revealed a reduced size of the skeletal elements and a reduced region of proliferating chondrocytes already at this early time. In addition, the hypertrophic region is significantly shorter in mutant radii ($p_+ < 0.001$) (Fig 5A and 5B). To test if chondrocyte proliferation is also affected prenatally, we labeled proliferating cells of E14.5 mice with BrdU. We found that compared to control littermates the proliferation rate of columnar chondrocyte of *Atoh8$^{flox/flox}$;Prx1-Cre$^{female}$* mice was reduced by 2.4% ($p_+ < 0.001$) (Fig 5C and 5D). As the initiation of hypertrophic differentiation requires a critical size of the skeletal element, the observed reduced hypertrophic region is likely a consequence of the reduced proliferation rate. In accordance, Alcian blue staining of micromass cultures revealed a reduced differentiation capacity of *Atoh8$^{flox/flox}$;Prx1-Cre$^{female}$* chondrocytes under differentiation conditions (Fig 5E). Since Col2 expression is generally decreased in Atoh8-deficient micromass cultures (Fig 1G), the reduced formation of cartilage nodules seems also be a consequence of decreased chondrocyte proliferation.

To investigate if the more severe phenotype in *Atoh8$^{flox/flox}$;Prx1-Cre$^{female}$* mice is based on defects in mesenchymal condensation, we analyzed cartilage anlagen of E12.5 limb buds, but could not detect differences between mutant and control mice (Fig 5F).

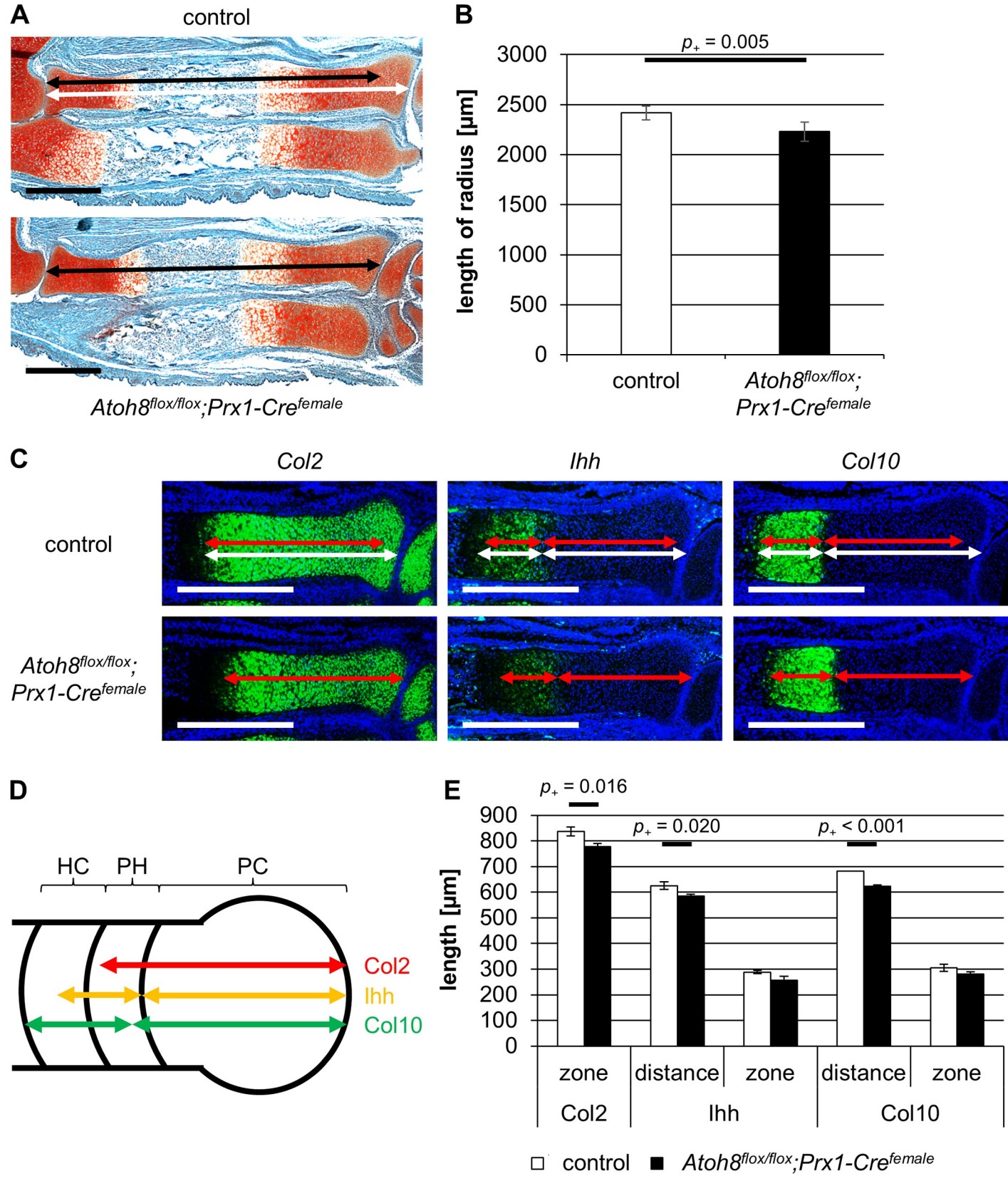

**Fig 4. *Atoh8^{flox/flox};Prx1-Cre^{female}* mice show a reduced skeletal size at prenatal stages.** (A, B) Sections of E16.5 control and *Atoh8^{flox/flox};Prx1-Cre^{female}* forelimbs stained with Safranin-Weigert. The radius of E16.5 *Atoh8^{flox/flox};Prx1-Cre^{female}* mutants is significantly shorter than that of control littermates. Measurements for B are visualized with white (control) and black (mutant) left-right arrows. (C-E) Parallel sections of E16.5 control and *Atoh8^{flox/flox};Prx1-Cre^{female}* forelimbs hybridized with a *Col2, Ihh and Col10* antisense probe show an accelerated hypertrophic differentiation in mutant mice. Measurements for E are visualized with white (control) and red (mutant) left-right arrows (C) and with red (Col2), yellow (Ihh) and green (Col10) left-right arrows in the schematic representation of the developing long bone (D). (D) PC = proliferating chondrocytes, PH = prehypertrophic chondrocytes, HC = hypertrophic chondrocytes. (B) n = 5 control and *Atoh8^{flox/flox};Prx1-Cre^{female}* mice from 2 litters; (E) n = 3 control and *Atoh8^{flox/flox};Prx1-Cre^{female}* mice from 1 litter; (B, E) Bayesian analysis. Scale bar in A (black lines) and C (white lines): 500 μm.

In summary, ubiquitous deletion of Atoh8 results in a reduced proliferation rate and consequently a delayed initiation of hypertrophic differentiation at early stages. At later stages the proliferation rate remains reduced, while the switch from proliferation to hypertrophy is accelerated leading to reduced regions of proliferating chondrocytes. Since the condensation of mesenchymal cells is not obviously affected, the increased severity of the phenotype compared to that of *Atoh8^{flox/flox};Col2a1-Cre* mice is likely due to a temporal delay in Atoh8 inactivation by the Col2a1 driven recombination.

## Calcineurin does not regulate nuclear localization of Atoh8 in chondrocytes

The calcium- and calmodulin-dependent serine/threonine protein phosphatase Calcineurin has been identified as an interaction partner of Atoh8 regulating its nuclear localization [30]. The authors showed that in HEK293 cells Calcineurin inhibition by Cyclosporine A (CsA) prevents the translocation of Atoh8 from the cytoplasm to the nucleus [30]. Since Calcineurin is known to regulate chondrogenesis [33], we asked if Calcineurin also controls the nuclear localization of Atoh8 in chondrocytes. For this purpose, we overexpressed an Atoh8-ECFP fusion protein in the chondrogenic cell line ATDC5 [29] and treated the cells with different concentrations of CsA. Fluorescence microscopy revealed that Atoh8-ECFP is primarily located in the nucleus in untreated cells (Fig 6A). Interestingly, while treatment with CsA clearly reduced nodule formation in micromass cultures as published previously (Fig 6B) [34], we could not detect an increased cytoplasmic localization of Atoh8 due to Calcineurin inhibition in ATDC5 cells (Fig 6A). In conclusion, our results indicate that in contrast to HEK293 cells the cellular localization of Atoh8 is not regulated by Calcineurin in chondrocytes.

## Atoh8 regulates onset of hypertrophic differentiation upstream and chondrocyte proliferation independent or downstream of Ihh signaling

Since Atoh8 shows an increased expression in prehypertrophic chondrocytes where the expression of Ihh is initiated (Fig 1D) and both, the region of proliferating chondrocytes and the proliferation rate, are regulated by Ihh signaling [2, 3], we analyzed a potential interaction of Atoh8 and Ihh signaling. To activate Ihh signaling, we treated limb explant cultures of E16.5 control and *Atoh8^{flox/flox};Prx1-Cre^{female}* mice with the Smoothened agonist Purmorphamine. We used *in situ* hybridization with a *Col2* antisense probe to identify the domain of proliferating chondrocytes, and BrdU incorporation to determine the proliferation rate. As expected, the size of the Col2 expression domain was significantly increased by Purmorphamine treatment in control mice ($p. < 0.001$). Similarly, the region of proliferating chondrocytes was significantly enlarged in Purmorphamine-treated *Atoh8^{flox/flox};Prx1-Cre^{female}* limb explants ($p. < 0.001$) (Fig 7A and 7B). In contrast, the effect of the Purmorphamine treatment on chondrocyte proliferation was significantly reduced in *Atoh8^{flox/flox};Prx1-Cre^{female}* compared to control limb explants. While the activation of Ihh signaling caused an upregulation of the proliferation rate by 1.1% in round ($p. < 0.001$) and by 3.2% in columnar ($p. < 0.001$) chondrocytes of

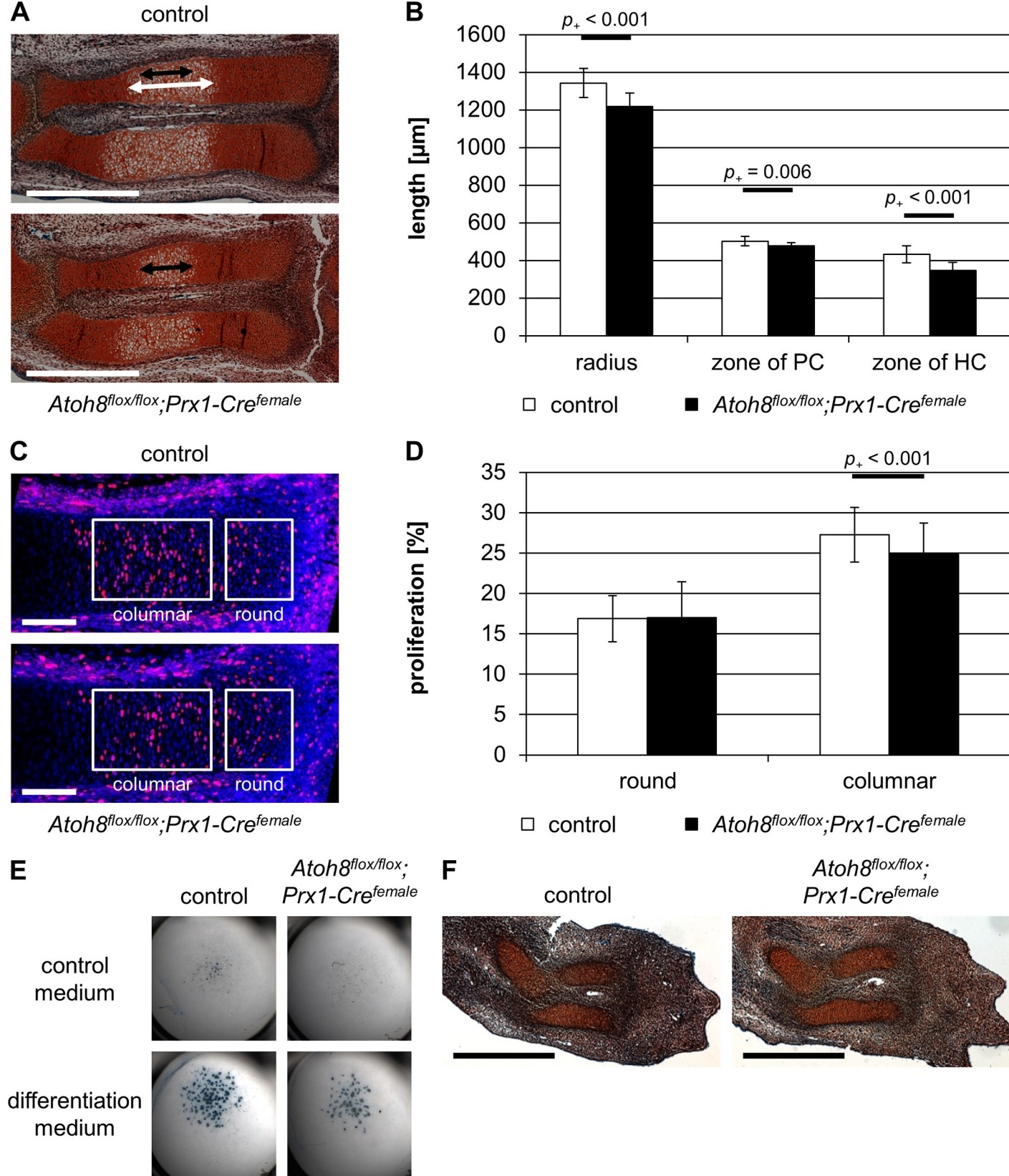

**Fig 5. Prenatal *Atoh8^flox/flox*;*Prx1-Cre^female* mice show reduced chondrocyte proliferation and defects in hypertrophic differentiation.** (A, B) Sections of E14.5 control and *Atoh8^flox/flox*;*Prx1-Cre^female* mutant forelimbs stained with Safranin-Weigert. The length of the zone of hypertrophic chondrocytes is reduced in *Atoh8^flox/flox*; *Prx1-Cre^female* mutant mice. Measurements for B are visualized with white (control) and black (mutant) left-right arrows. HC = hypertrophic chondrocytes; PC = proliferating chondrocytes. (C, D) BrdU labeling of E14.5 control and *Atoh8^flox/flox*;*Prx1-Cre* forelimb sections revealed a reduced proliferation rate in columnar chondrocytes in *Atoh8^flox/flox*;*Prx1-Cre^female* mice. Regions of round and columnar chondrocytes analyzed in D are framed with white boxes. (E) Alcian blue staining of chondrocytes cultured in high-density for 14 days in control or differentiation medium. Primary chondrocytes were isolated from E12.5 control and *Atoh8^flox/flox*; *Prx1-Cre^female* mutant embryos. The number of cartilage nodules is decreased in micromass cultures of mutant mice. (F) Sections of E12.5 control and *Atoh8^flox/flox*; *Prx1-Cre^female* forelimbs stained with Safranin-Weigert do not show obvious differences in size of cartilage anlagen. (B) n = 18 control and 17 *Atoh8^flox/flox*;*Prx1-Cre^female* mice from 8 litters; (D) n = 12 control and 11 *Atoh8^flox/flox*;*Prx1-Cre^female* biological replicates from 5 litters; (B, D) Bayesian analysis. Scale bar in A: 500 μm and in C: 100 μm (white lines), in F: 500 μm (black lines).

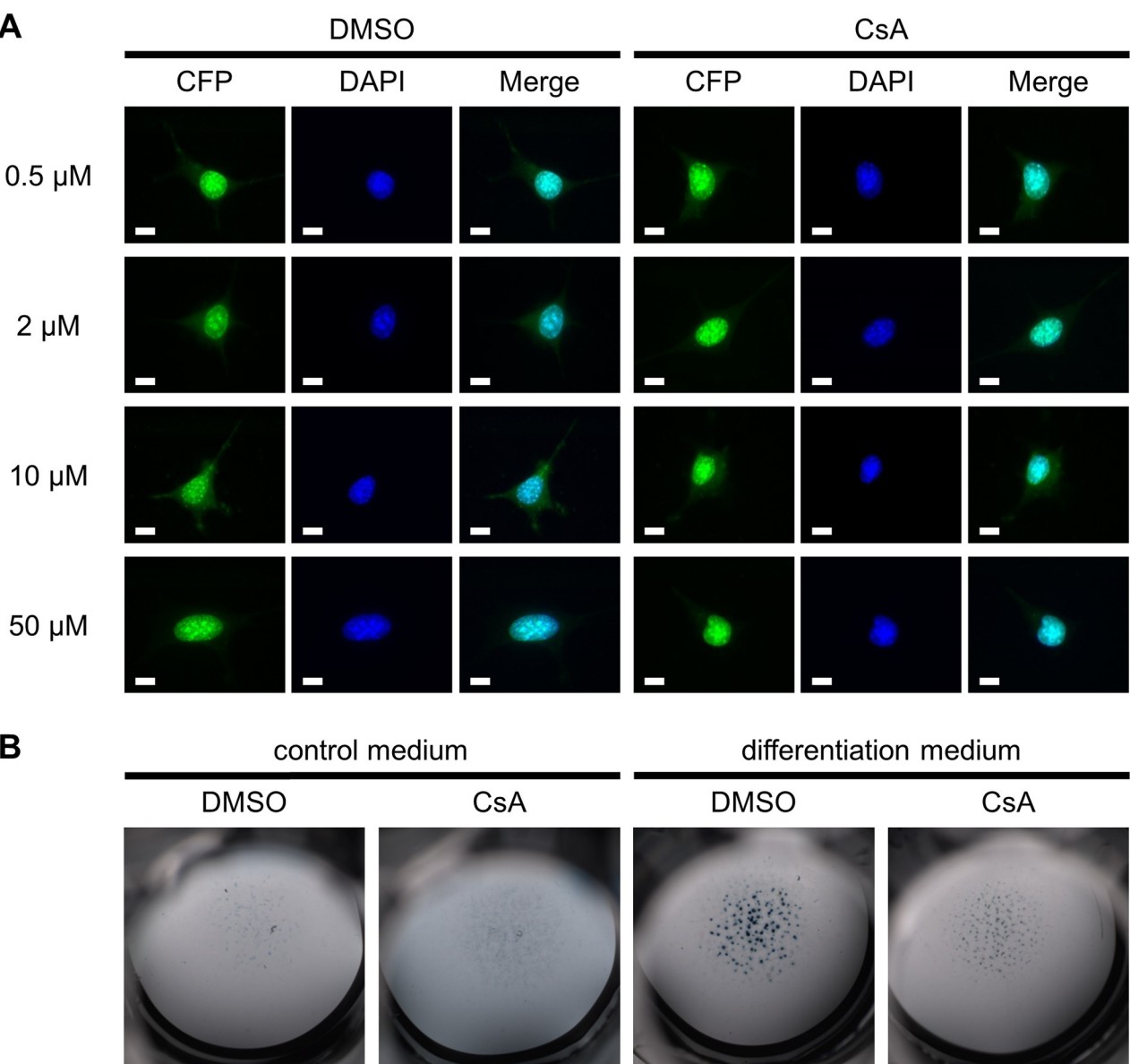

**Fig 6. The nuclear localization of Atoh8 is Calcineurin-independent in chondrocytes.** (A) Fluorescence images of hAtoh8-CFP overexpressing ATDC5 cells treated with different concentration of CsA for 24 h. CsA treatment does not change the nuclear localization of Atoh8. Scale bar (white lines): 10 μm. (B) Alcian blue staining of chondrocytes cultured in high-density for 14 days in control or differentiation medium. Primary chondrocytes were isolated from E12.5 wild-type embryos and treated with 10 μM CsA or DMSO as a control. The number of cartilage nodules is decreased in micromass cultures treated with CsA.

control embryos, it did not or only slightly increase the proliferation rate of round ($p_-$ = 0.397) and columnar chondrocytes ($p_-$ = 0.010) in *Atoh8^flox/flox^;Prx1-Cre^female^* limb explants (Fig 7C and 7D). These results indicate that Atoh8 acts independent or downstream of Ihh signaling in determining the rate of chondrocyte proliferation. In contrast, the size of the domain of proliferating chondrocytes seems to be regulated by Ihh downstream of Atoh8. This was supported by qRT-PCR analyses of E14.5 and E16.5 forelimb skeletal elements revealing a significantly reduced Ihh expression in *Atoh8^flox/flox^;Prx1-Cre^female^* mutants at both stages ($p_+$ = 0.014 and $p_+$ = 0.008) (Fig 7E). Consistently, the Ihh signal detected by *in situ* hybridization of E16.5 embryos seems to be weaker in *Atoh8^flox/flox^;Prx1-Cre^female^* mice compared to control littermates (Fig 4C). In summary, Atoh8 regulates the onset of hypertrophic differentiation upstream of Ihh, while it has an Ihh independent effect or acts downstream of Ihh on the proliferation rate.

## Discussion

Members of the *Atonal* gene family, e.g. *Atoh1* and *Atoh7*, have been associated with the development of sensory neurons and are typically classified as proneuronal genes [35–37]. Here we have investigated the role of Atoh8, a less well characterized member of the protein family, which is widely expressed in the developing embryo [8, 9, 14, 38, 39].

The general role of Atoh8 during embryonic development has been controversially discussed, since the phenotypic spectrum of mouse mutants reaches from early embryonic lethality [39] to no severe phenotypical alterations in an independently derived loss of function mouse line [40]. A mild phenotype has also been described after pancreas-specific inactivation of a conditional Atoh8 allele [41]. In this study, we have used an independently generated conditional Atoh8 allele to induce a chondrocyte-specific and a ubiquitous inactivation of Atoh8 in mice. Both mutants show a reduced skeletal size, but are viable and survive until adulthood. Since no other *Atoh* gene is expressed in control or Atoh8-deficient developing skeletal elements, it can be excluded that the bone phenotype of Atoh8-deficient mice is diminished due to a functional redundancy or compensation by other Atoh proteins. This strongly supports a role of Atoh8 in fine tuning developmental processes rather than being required for survival.

Detailed analysis revealed a decreased chondrocyte proliferation rate, which is likely responsible for the decreased bone length. Furthermore, both mutant lines show reduced regions of proliferating and hypertrophic chondrocytes indicating an additional role of Atoh8 in controlling the differentiation process. To exclude that the reduced skeletal size is due to defects in the condensation we analyzed mice at E12.5, but could not detect any differences in skeletal size. Consistent with this, deleting Atoh8 under the Col2-promoter, which is active after the chondrocyte fate is established, also reduces the proliferation rate and accelerates hypertrophic differentiation strongly indicating a direct role of Atoh8 in both processes.

Corresponding to our results, Atoh8 has been identified as a regulator of proliferation and differentiation in other tissues. Nevertheless, its function seems to be quite variable depending on the tissue type. For instance, in endothelial cells and podocytes Atoh8 has been identified as a negative regulator of proliferation [9, 42] while in chondrocytes (this study), in regenerating mouse myoblasts [10] and in colorectal cancer Atoh8 [11] seems to activate the proliferation rate. Different sets of interacting factors are thus likely to determine the cell type specific function of Atoh8 as a regulator of proliferation.

Although little is known about the proteins that interact with Atoh8, the serine/threonine protein phosphatase Calcineurin has recently been shown to regulate its nuclear localization in HEK293 cells. While in untreated cells Atoh8 was mainly located in the nucleus, inhibition of

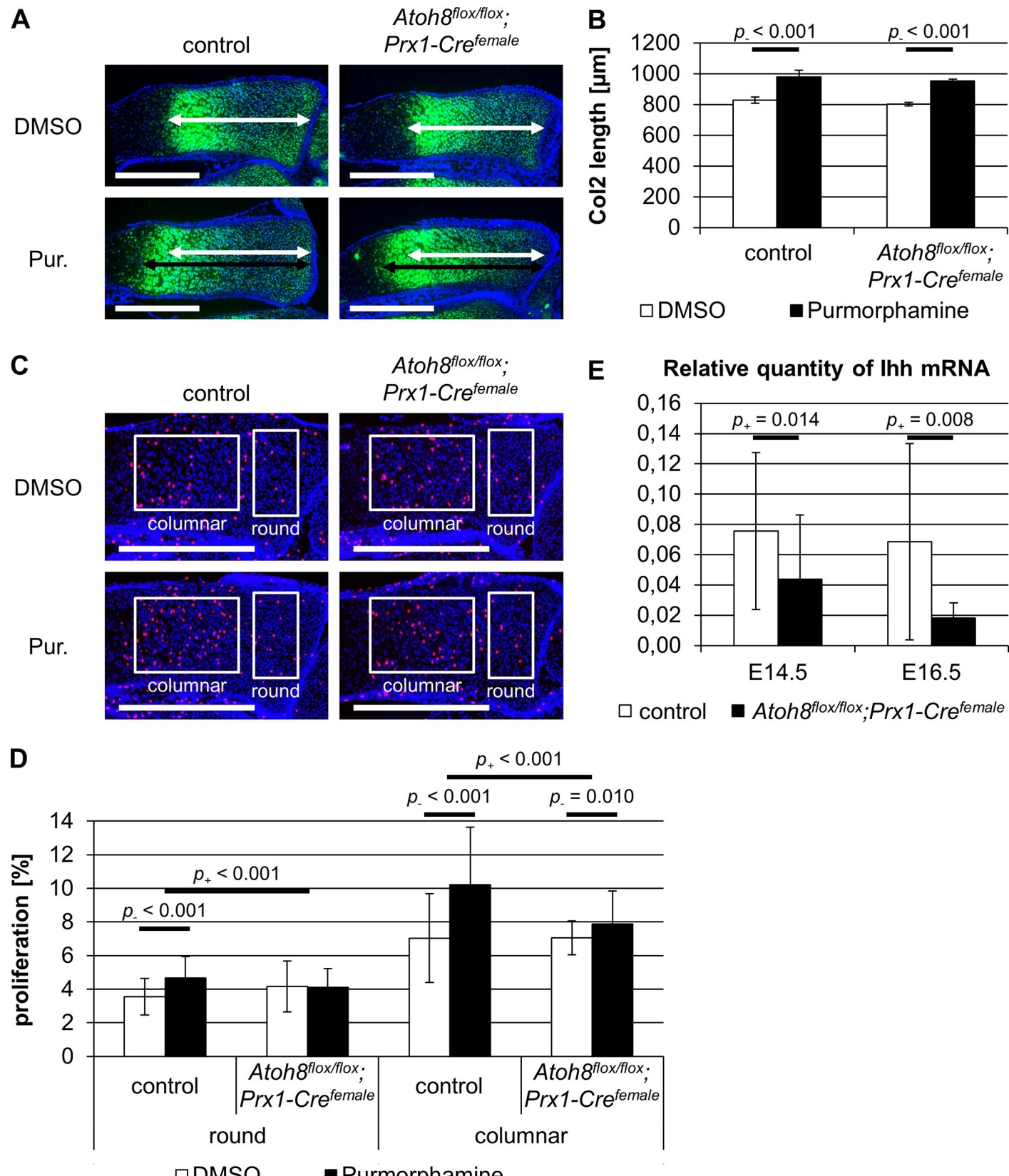

**Fig 7. Atoh8 seems to regulate chondrocyte proliferation in parallel or downstream but hypertrophic differentiation upstream of Ihh signaling.** (A, B) Sections of limb explants hybridized with a *Col2* antisense probe. Limb explants of E16.5 control and *Atoh8^{flox/flox};Prx1-Cre^{female}* forelimbs were cultured for 96 h with 4 μM Purmorphamine (Pur.) or DMSO as a control. Measurements for B are visualized with white (untreated) and black (treated) left-right arrows. The zone of proliferating chondrocyte is enlarged in control and *Atoh8^{flox/flox};Prx1-Cre^{female}* radii after Purmorphamine treatment. (C, D) BrdU labeling of E16.5 control and *Atoh8^{flox/flox};Prx1-Cre^{female}* limb explants, which were cultured for 48 h with 4 μM Purmorphamine (Pur.) or DMSO as a control. Regions of round and columnar chondrocytes analyzed in D are framed with white boxes. Activation of Ihh signaling by Purmorphamine increases chondrocyte proliferation in control limb explants more than in *Atoh8^{flox/flox};Prx1-Cre^{female}* explants. (E) qRT-PCR of E14.5 and E16.5 control and *Atoh8^{flox/flox};Prx1-Cre^{female}* forelimb skeletal elements revealed reduced Ihh expression in mutant mice at both stages. (B) n = 2 control and *Atoh8^{flox/flox};Prx1-Cre^{female}* mice from 1 litter; (D) n = 9 control and 8 *Atoh8^{flox/flox}; Prx1-Cre^{female}* mice from 4 litters; (E) n = 12 control and 10 *Atoh8^{flox/flox};Prx1-Cre^{female}* mice from 6 litters (E14.5), n = 9 control and 10 *Atoh8^{flox/flox};Prx1-Cre^{female}* mice from 5 litters (E16.5); (B, D, E) Bayesian analysis. Scale bar (white lines) in A and C: 500 μm.

Calcineurin by CsA led to the retention of Atoh8 in the cytoplasm [30]. Interestingly, Calcineurin has also been identified as a regulator of chondrocyte differentiation [33]. In chondrocytes Calcineurin controls the nuclear localization of Nfat transcription factors and consequently the expression of chondrocyte-specific regulators including Sox9, Bmp2, Fgf18 [34, 43, 44] and a large number of extracellular matrix components like Acan, Col2a1, Col9a1, Col11a1 and Col10a1 [45, 46]. To test if Calcineurin regulates the cellular localization of Atoh8 in chondrocytes, we inhibited its function by CsA treatment in the chondrogenic cell line ATDC5, but could not detect any indication of an increased cytoplasmic Atoh8 localization. The dependency of nuclear Atoh8 localization on Calcineurin activity seems thus be specific for other cell types.

To integrate Atoh8 into the regulatory network of chondrocytes, we next investigated a potential interaction with Ihh signaling, which regulates the proliferation rate and the pool of proliferating chondrocytes [2, 3, 47]. Ihh is expressed in prehypertrophic chondrocytes and its expression domain overlaps with the region of increased Atho8 expression at the border between proliferating and hypertrophic chondrocytes. While chondrocyte proliferation is directly regulated by Ihh secreted from prehypertrophic cells [48, 49], the pool of proliferating chondrocytes is regulated indirectly by the Ihh-dependent expression of parathyroid hormone related peptide (PTHrP) in round chondrocytes. PTHrP in turn acts back on proliferating chondrocytes, where it prevents the onset of hypertrophy [5, 50]. Interestingly, the expression of Ihh detected by *in situ* hybridization and qRT-PCR and, subsequently, the region of proliferating chondrocytes is reduced in Atoh8 mutants indicating that the Ihh/PthrP system acts downstream of Atoh8 on the onset of differentiation. This is supported by limb explant cultures treated with the hedgehog agonist Purmorphamine, which rescued the reduced region of proliferating cells in Atoh8-deficient explants. In contrast, activation of Ihh signaling by Purmorphamine did not fully rescue the proliferation rate in limb explant cultures of Atoh8-deficient mice strongly supporting an Ihh independent role of Atoh8 in regulating chondrocyte proliferation. If Atoh8 really acts in parallel or alternatively downstream of Ihh has to be investigated in future studies.

Similar to Atoh8, Fibroblast growth factor (Fgf) and Bone morphogenetic protein (Bmp) signaling have been shown to regulate chondrocyte proliferation independent of Ihh, whereas they modulate the onset of hypertrophy by affecting Ihh expression. While Bmp signaling induces proliferation and Ihh expression, Fgf signals inhibit both processes [51, 52]. Atoh8 might thus act downstream of either signaling system or act on proliferation and differentiation as an independent regulator downstream of yet not identified mechanisms.

With respect to a potential interaction of Ihh signaling and Atoh8 function it is interesting to note that E-box elements, the typical target sequence of bHLH proteins, have been identified as unusual targets of Gli transcription factors [53], the main mediators of Ihh signaling. Accordingly, Atoh8 and Gli might interact at these elements to regulate target gene expression. In preliminary co-immunoprecipitation experiments we could, however, not detect an interaction of Atoh8 and Gli3 (unpublished). Nevertheless, the proteins might interact in larger complexes or

competitively bind to these target sequences. Alternatively, Atoh1 has been reported to regulate the transcription Cep131 and thereby influences the maintenance of the primary cilium, an essential organelle for Ihh signal transduction [54]. Aoth8 might have a similar function and regulate Ihh/Gli3 via modification of cilia structure.

In conclusion, we identified Atoh8 as a regulator of endochondral ossification, which increases the pool of proliferating chondrocytes by activating the rate of chondrocyte proliferation and inhibiting the onset of hypertrophic differentiation. Our data further demonstrate that Atoh8 acts upstream and in parallel or downstream of Ihh in regulating the onset of hypertrophy and the proliferation rate, respectively. Atoh8 thus represents a new factor controlling the tight balance between proliferation, cell cycle exit and hypertrophy, a critical step to coordinate bone growth.

## Supporting information

**S1 Fig. Targeting strategy for Atoh8 deletion in mice.** Schematic representation of wild-type (A), targeted (B) and recombined (C) *Atoh8* gene locus. Exon 1 is flanked by two *loxP* sites [17]. Genotyping primers are shown as black arrows, the length of corresponding PCR products in bp are listed in between.
(PDF)

**S2 Fig. Comparison of control, heterozygous and homozygous Atoh8 deleted mice.** (A) Radius length of P7 control, *Atoh8$^{flox/+}$;Col2a1-Cre* and *Atoh8$^{flox/flox}$;Col2a1-Cre* mice. (B, C) Radius length of E14.5 (B) and E16.5 (C) control, *Atoh8$^{flox/+}$;Prx1-Cre$^{female}$* and *Atoh8$^{flox/flox}$; Prx1-Cre$^{female}$* mice. (A) n = 10 control, 6 *Atoh8$^{flox/+}$;Col2a1-Cre* and 8 *Atoh8$^{flox/flox}$;Col2a1-Cre* mice from 5 litters; (B) n = 18 control, 8 *Atoh8$^{flox/+}$;Col2a1-Cre* and 17 *Atoh8$^{flox/flox}$;Prx1-Cre$^{female}$* mice from 8 litters; (C) n = 5 control, 3 *Atoh8$^{flox/+}$;Col2a1-Cre* and 5 *Atoh8$^{flox/flox}$;Prx1-Cre$^{female}$* mice from 2 litters; Bayesian analysis.
(PDF)

**S3 Fig. Radius length of prenatal *Atoh8$^{flox,flox}$;Col2a1-Cre* mice is not reduced.** Radius length of E16.5 control and *Atoh8$^{flox/flox}$;Col2a1-Cre* mice are comparable. n = 2 control and *Atoh8$^{flox/flox}$;Col2a1-Cre* mice from 2 litters; Bayesian analysis.
(PDF)

**S4 Fig. *Atoh8* is the only *Atoh* gene which is noteworthy expressed in chondrocytes.** Relative Atoh1 (A), Atoh2 (B), Atoh3 (C), Atoh4 (D), Atoh5 (E), Atoh7 (F) and Atoh8 (G) mRNA expression of NMRI embryos (grey) or control (white) and *Atoh8$^{flox/+}$;Prx1-Cre$^{female}$* (black) forelimb skeletal elements. n = 2 wild-type mice from 2 litters; n = 4 control and 3 *Atoh8$^{flox/+}$; Prx1-Cre$^{female}$* mice from 2 litters.
(PDF)

**S1 Table. Overview about calculated $p_-$ and $p_+$ values.** The probability of a negative effect ($p_-$) and a positive effect ($p_+$) caused by an Atoh8 deletion or a Purmorphamine treatment were determined by Bayesian analysis. All calculated $p_-$ and $p_+$ values for the different experiments are listed in this table.
(PDF)

**S2 Table. Overview of additional primer pairs used for the expression analysis of the different *Atoh* genes by qRT-PCR.**
(PDF)

## Acknowledgments

We would like to thank Sabine Schneider for excellent technical assistance with limb explant preparations.

## Author Contributions

**Conceptualization:** Manuela Wuelling, Beate Brand-Saberi, Andrea Vortkamp.

**Formal analysis:** Daniel Hoffmann.

**Funding acquisition:** Manuela Wuelling, Beate Brand-Saberi, Andrea Vortkamp.

**Investigation:** Nadine Schroeder.

**Methodology:** Nadine Schroeder, Manuela Wuelling, Andrea Vortkamp.

**Project administration:** Nadine Schroeder, Manuela Wuelling.

**Resources:** Beate Brand-Saberi.

**Supervision:** Andrea Vortkamp.

**Validation:** Nadine Schroeder, Manuela Wuelling, Andrea Vortkamp.

**Visualization:** Nadine Schroeder.

**Writing – original draft:** Nadine Schroeder, Andrea Vortkamp.

**Writing – review & editing:** Nadine Schroeder, Manuela Wuelling, Andrea Vortkamp.

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
