## [Decision Letter · Decision Letter 0]

20 Jun 2019

PONE-D-19-14765

Atoh8 acts as a regulator of chondrocyte proliferation and differentiation in endochondral bones

PLOS ONE

Dear Dr. Vortkamp,

Thank you for submitting your manuscript to PLOS ONE. After careful consideration, we feel that it has merit but does not fully meet PLOS ONE’s publication criteria as it currently stands. Therefore, we invite you to submit a revised version of the manuscript that comprehensively addresses the points raised by both reviewers.

We would appreciate receiving your revised manuscript by Aug 04 2019 11:59PM. To enhance the reproducibility of your results, we recommend that if applicable you deposit your laboratory protocols in protocols.io, where a protocol can be assigned its own identifier (DOI) such that it can be cited independently in the future. For instructions see: http://journals.plos.org/plosone/s/submission-guidelines#loc-laboratory-protocols

We look forward to receiving your revised manuscript.

Kind regards,

Andre van Wijnen

Academic Editor

PLOS ONE

Journal Requirements:

Reviewers' comments:

Reviewer's Responses to Questions

**Comments to the Author**

1. Is the manuscript technically sound, and do the data support the conclusions?

Reviewer #1: Partly

Reviewer #2: Partly

2. Has the statistical analysis been performed appropriately and rigorously? 

Reviewer #1: Yes

Reviewer #2: I Don't Know

3. Have the authors made all data underlying the findings in their manuscript fully available?

Reviewer #1: Yes

Reviewer #2: Yes

4. Is the manuscript presented in an intelligible fashion and written in standard English?

Reviewer #1: Yes

Reviewer #2: Yes

5. Review Comments to the Author

Reviewer #1: Atoh8 is a broadly expressed transcription factor affecting differentiation in many organs. Its expression in avian growth plate and mouse limb buds prompted the present study of any role in limb development.

The investigators have knocked out Atoh8 using both Prx-Cre for germline and Col2a1-Cre for chondrocytes. The stages of phenotype development were different for the two. Zones of proliferative and hypertrophic chondrocytes were reduced for both, but the chondrocyte-specific knockouts had shortened bones postnatally whereas the Prx-Cre mice did so already at birth. Evidence was presented of interaction of Atoh8 with IHH signalling and expression levels.

1. The paper is descriptive only , revealing a contributory role for Atoh8 in limb endochondral bone formation but no mechanistic insights. In its technical aspects the work has been carried out to a high standard.

2. The insertion of figure legends in the text is unusual. Maybe it is a policy of the journal? It is a distraction, interrupts the flow of description of Results.

3. Figure 1 needs a better description. Its details and interpretation would not necessarily be obvious to readers of the journal. The zones of limb buds need to be specified clearly in the figure. It is stated that Atoh8 is expressed in the proliferating and hypertrophic zones but especially in prehypertrophic, but it is not so clear in 1C that Atoh8 is expressed in the hypertrophic zone (Col X). Another interpretation is that Atoh8 is not expressed in hypertrophic but appears to be so in prehypertrophic.

4. Fig 1D and E. What do the micromass culture data mean, apart from showing the knockdown of Atoh8?

5. Fig 4C and P 13 lines 366 -373. This explanation of Fig 4C is difficult to understand, especially lines 369-371. The authors might explain carefully just exactly how these measurements were made.

6. Fig 4C. It is impossible to discern any detail in 4C, to the extent that there seems little point in including it iof better images are not available. . This is particularly so in view of the extremely small . although statistically significant, difference between columnar cells of control and Atoh-deficient (Fig 5D).

7. Fig 6 and Discussion, lines 501-510. The negative outcome of this experiment is not conclusive. Atoh8 is located primarily in the nucleus and cannot be detected in cytoplasm with Csa treatment. Excluding the possibility of Csa outside the nucleus would require more throrough investigation, rather than concluding as has been done in Discussion lines 501 et seq.

8. Fig 7 . The quantitative data in Fig 7 are unconvincing, and in 7C the same comment is made as for Fig 4C (above) – it is impossible to discern any detail in 7C. Furthermore the quantitative data is so weak that it is difficult to accept the conclusions that Atoh 8 acts upstream andin parallel or downstream of Ihh in regulating the onset of hypertrophy and proliferation rate, respectively ( Discussion lines 547-550). Indeed the linkage between Ihh and PTHrP is so strong physiologically that any attempt to reach conclusions about Atoh8 - Ihh relationship in isolation is beset with difficulty.

Reviewer #2: The aim of this study is to uncover the role of atonal homolog 8 (Atoh8) transcription factor in endochondral skeletal growth. The authors first studied expression of Atoh8 in developing limbs by whole mount ISH and conventional ISH on tissue sections, and the qPCR analysis of RNA prepared from micromass cultures of control and Atoh8 CKO. The authors then carried out histological and morphometric analyses of various parameters of control (Cre negative litter mates) and conditional Atoh8 deficient mutant samples (either Col2a1Cre, Atoh8 floxed or Prx1-Cre, Atoh8floxed mice). Expression analysis revealed that Atoh8 was expressed in cartilage in developing skeleton. In addition, ablation of the gene resulted in marked reduction in collagen 2 expression at least in cultured chondrocytes. In contrast, histological and various morphometric analyses of the mutant mice showed very modest differences compared to those in control mice. Though the authors claim the possible role of Atoh8 in regulation of skeletal growth and interaction with Ihh signaling, the reviewer thinks that the study is inconclusive due to a small size of sample number and insufficient information on the mouse strains. They should also consider functional redundancy among other family members and more carefully interpret the results. Otherwise, their conclusion and discussion potentially mislead the readers.

The culturs prepared from E12.5 limb buds contain various cell populations. The results from this culture system should not be interpreted as the phenotype of chondrocytes.

To identify and establish very modest differences in size of skeleton and histomorphometric parameter of growth-plate, the authors need to have appropriate control and analyze enough number of specimens. In this study, the authors always used Cre negative littermates as control, but the strains of the parent mice are not given. If the compound mice generated in this study came from more than 2 strains, they need to consider impact of genetic drift on skeletal growth difference. Col2Cre could be an another control mouse. The sample size for analysis of skeleton length is 3-4, which may not be enough to conclude the growth retardation in the mutant mice. It is not provided how many biological different samples were used for histological analysis although the sample size ranges 10-12?. It should be provided how they define the anatomical landmark to orient the limbs and to secure the longest or same tissue level for measurement of various histological parameters.

By searching “GenePaint” database, the reviewer found other members of Atoh families are also expressed in developing skeleton. Are there functional redundancy among other family members? Have you checked expression of other Atoh transcription factors? These points are required to reach conclusion on the exclusive role of Atoh 8 in endochondral ossification.

6. PLOS authors have the option to publish the peer review history of their article (what does this mean?). If published, this will include your full peer review and any attached files.

Reviewer #1: No

Reviewer #2: No

---

## [Author Response · Author response to Decision Letter 0]

1 Aug 2019

Response to reviewer #1:

1. The paper is descriptive only, revealing a contributory role for Atoh8 in limb endochondral bone formation but no mechanistic insights. In its technical aspects the work has been carried out to a high standard.

In our publication, we initially describe the bone phenotype of mice with a chondrocyte-specific or a ubiquitous Atoh8 deletion and we show that Atoh8 regulates endochondral bone growth by activating chondrocyte proliferation and inhibiting chondrocyte differentiation. Additionally, functional data from in vivo and in vitro experiments indicates that Atoh8 regulates Ihh expression and in this way modulates the onset of differentiation, while it regulates chondrocyte proliferation either downstream or independent of Ihh. Although the data includes a morphological analysis, it clearly reveals first mechanistic insights into the interaction of Ihh and the transcription factor Atoh8.

2. The insertion of figure legends in the text is unusual. Maybe it is a policy of the journal? It is a distraction, interrupts the flow of description of Results.

The insertion of figure legends in the text corresponds to the guidelines of PLOS ONE.

3. Figure 1 needs a better description. Its details and interpretation would not necessarily be obvious to readers of the journal. The zones of limb buds need to be specified clearly in the figure. It is stated that Atoh8 is expressed in the proliferating and hypertrophic zones but especially in prehypertrophic, but it is not so clear in 1C that Atoh8 is expressed in the hypertrophic zone (Col X). Another interpretation is that Atoh8 is not expressed in hypertrophic but appears to be so in prehypertrophic.

For a better visualization of the data summarized in Figure 1, we included a whole-mount in situ hybridization using a Col2 probe (1C, revised version) to demarcate the cartilage templates in the limb buds. Moreover, we added a schematic model showing the zones of proliferating, prehypertrophic and hypertrophic chondrocytes as well as the Col2 and Col10 expression domains (1E, revised version). Atoh8 can be detected in all three chondrocyte populations with stronger expression in prehypertrophic chondrocytes. The hybridization signal is clearly visible in the high-resolution images of the manuscript (tiff file Fig. 1C in the first version or Fig. 1D in the revised version). 

4. Fig 1D and E. What do the micromass culture data mean, apart from showing the knockdown of Atoh8?

As the reviewer states correctly, the verification of the Atoh8 knockdown is one important reason for showing the qPCR data obtained from micromass cultures (Figure 1D and E in the first version or Fig. 1F and G in the revised version). In addition, these data show that Atoh8 expression is maintained during the differentiation process in vitro. Although the expression is very low at d14, it increases again at later stages. This is in line with the results of the Atoh8 in situ hybridization revealing Atoh8 expression in proliferating, prehypertrophic and hypertrophic chondrocytes and with the observation that Atoh8 regulates not only the onset, but also the process of hypertrophic differentiation. Furthermore, the decreased Col2 expression in Atoh8-deficient micromass cultures indicates that the reduced nodule formation is at least partly based on the decreased proliferation of Atoh8-deficient chondrocytes as mentioned on p. 15 lines 385-387 (first version).

5. Fig 4C and P 13 lines 366 -373. This explanation of Fig 4C is difficult to understand, especially lines 369-371. The authors might explain carefully just exactly how these measurements were made.

For a better representation of our data, we added a schematic representation of the analyzed bone showing the zones of proliferating, prehypertrophic and hypertrophic chondrocytes and the Col2, Ihh and Col10 expression domains in Figure 4 (4D, revised version). We also included a detailed description of how we performed the measurements in the materials and methods section. 

6. Fig 4C. It is impossible to discern any detail in 4C, to the extent that there seems little point in including it if better images are not available. This is particularly so in view of the extremely small. Although statistically significant, difference between columnar cells of control and Atoh-deficient (Fig 5D).

We agree that it is difficult to discern details in Fig. 5C in the PDF file generated upon submission. However, in the original high-resolution image submitted (tiff file) the labeled cells are clearly visible. As the reviewer states correctly, the differences between mutant and control mice are only small at this developmental stage (around 2.5 %). Nevertheless, a slight, but continuously reduced proliferation rate will ultimately lead to a reduced skeletal size. 

7. Fig 6 and Discussion, lines 501-510. The negative outcome of this experiment is not conclusive. Atoh8 is located primarily in the nucleus and cannot be detected in cytoplasm with Csa treatment. Excluding the possibility of Csa outside the nucleus would require more throrough investigation, rather than concluding as has been done in Discussion lines 501 et seq.

It has been shown before that the nuclear Atoh8 translocates to the cytoplasm upon CsA treatment in HEK cells (Chen et al., 2016). In contrast, the results presented here demonstrate that under the same experimental conditions (the same CsA and the same Atoh8-Flag expression construct as used in Chen et al., 2016) the nuclear localization of Atoh8 is not altered by CsA treatment in ATDC5 cells indicating a cell type specific regulation of Atoh8 by Calcineurin. In fact, the potential regulation of Atoh8 by CsA and the corresponding connection to NFAT was one of the reasons we decided to investigate Atoh8 function in chondrocytes. As the reviewer states the result was negative concerning the localization dependency of Atoh8 on Calcineurin. Nevertheless, we thought it is important to publish the cell type specific regulation of Atoh8 localization, while the localization and activity of Calcineurin was not the scope of this study. 

8. Fig 7. The quantitative data in Fig 7 are unconvincing, and in 7C the same comment is made as for Fig 4C (above) – it is impossible to discern any detail in 7C. Furthermore the quantitative data is so weak that it is difficult to accept the conclusions that Atoh 8 acts upstream and in parallel or downstream of Ihh in regulating the onset of hypertrophy and proliferation rate, respectively (Discussion lines 547-550). Indeed the linkage between Ihh and PTHrP is so strong physiologically that any attempt to reach conclusions about Atoh8 - Ihh relationship in isolation is beset with difficulty.

As discussed for figure 5C, the low quality of Fig. 7C is based on the conversion of a high-resolution tiff file visualizing details to a PDF file. 

We fully agree that the interaction of Ihh and Pthrp is complex. While Ihh regulates chondrocyte proliferation independent of Pthrp, the onset of hypertrophic differentiation/pool of proliferating chondrocytes is regulated by the upregulation of Pthrp. Importantly, the expression of Pthrp is strongly dependent on Ihh and reduced Ihh signaling or activation by Purmorphamine treatment is followed by the respective alterations in the Pthrp levels (Han et al., 2016). 

Our hypothesis that Atoh8 differentially interacts with Ihh in regulating chondrocyte differentiation and hypertrophy is based on several lines of evidence: The data in Fig. 7E show that the Ihh expression is significantly decreased in Atoh8 mutants (around 40 % at E14.5 and around 70 % at E16.5) indicating that Atoh8 activates Ihh expression. The reduced Ihh expression is linked to an accelerated onset of hypertrophic differentiation as shown in Fig. 3D and 4E placing Atoh8 upstream of the Ihh/Pthrp system. This is supported by data from the limb explant cultures (Fig. 7A and B) demonstrating that activation of the Ihh/PthrP system by Purmorphamine increases the pool of proliferating chondrocytes independent of Atoh8 function. 

Reduced Ihh levels will also lead to a reduced chondrocyte proliferation rate. However, the data shown in Fig. 7C and D indicate that activation of Ihh signaling does not rescue the proliferation rate downstream of Atoh8 strongly pointing to an Ihh independent or downstream function of Atoh8. A similar relationship has also been observed for FGF signaling which acts upstream of Ihh on the differentiation (inhibits Ihh expression) and independent or downstream on the proliferation rate (Minina et al. 2002). 

In order to clarify the interdependency of Ihh and Pthrp, we changed the discussion in lines 516-528 (first version, see lines 548-561 in the revised version).

Response to reviewer #2:

1. The cultures prepared from E12.5 limb buds contain various cell populations. The results from this culture system should not be interpreted as the phenotype of chondrocytes.

We agree that micromass cultures can only supplement morphological in vivo data. In this context, we used this culture system to complement the observations made in the Atoh8 mutants (Fig. 5). The reduced Col2 expression and the reduced nodule formation in Atoh8-deficient cultures are in line with the reduced chondrocyte proliferation and the defects in initial chondrocyte differentiation we found in Atoh8 mutant mice (see also point 4 reviewer 1).

2. To identify and establish very modest differences in size of skeleton and histomorphometric parameter of growth-plate, the authors need to have appropriate control and analyze enough number of specimens. In this study, the authors always used Cre negative littermates as control, but the strains of the parent mice are not given. If the compound mice generated in this study came from more than 2 strains, they need to consider impact of genetic drift on skeletal growth difference. Col2Cre could be another control mouse. The sample size for analysis of skeleton length is 3-4, which may not be enough to conclude the growth retardation in the mutant mice. It is not provided how many biological different samples were used for histological analysis although the sample size ranges 10-12? It should be provided how they define the anatomical landmark to orient the limbs and to secure the longest or same tissue level for measurement of various histological parameters.

As stated in our discussion Atoh8 deletion results in a modest bone phenotype, which due to the biological variability is difficult to analyze using conventional statistical analyses. Therefore, we implemented a Bayesian probability based statistical analysis, which takes differences between litters and experiments into account and allows to generate meaningful data with relatively low sample sizes and animal numbers. 

In addition, we added detailed information about the sample numbers (see figure legends), the numbers of sections analyzed and a more detailed description about how the measurements were performed (see lines 118, 131, 146). 

Both Cre lines have been used in multiple studies and no effect of the Cre allele has been published to our knowledge. To further exclude effects of the Cre driver line, we compared the bone length of mutant, heterozygous and control mice (Fig. S2). These data show, that the reduced bone length is based on the Atoh8 deletion and not on the Cre recombinase (Fig. S2). Moreover, we analyzed two mouse strains with different Cre recombinases at different stages in our study (Figures 2, 3, 4 and 5) and found comparable differences between mutants and controls making it unlikely that the effect observed is due to two different Cre alleles. 

To exclude strain differences all mouse lines were kept on a C57BL/6J background and mutants were compared to littermates.

3. By searching "GenePaint" database, the reviewer found other members of Atoh families are also expressed in developing skeleton. Are there functional redundancies among other family members? Have you checked expression of other Atoh transcription factors? These points are required to reach conclusion on the exclusive role of Atoh 8 in endochondral ossification.

As the reviewer mentioned, in situ hybridization data of sagittal E14.5 sections available at “GenePaint” indicate an expression of other Atoh genes in the developing skeleton. Based on this, we performed qPCR analyses to test the expression of Atoh1, Atoh2, Atoh3, Atoh4, Atoh5 and Atoh7 in forelimb skeletal elements of E14.5 control and Atoh8-deficient embryos. We could however not detect expression of any of these Atoh genes in control or Atoh8-deficient mice indicating that there is no functional redundancy or compensation during endochondral ossification. We added these results as Fig. S4 to the supplementary material.

---

## [Decision Letter · Decision Letter 1]

15 Aug 2019

Atoh8 acts as a regulator of chondrocyte proliferation and differentiation in endochondral bones

PONE-D-19-14765R1

Dear Dr. Vortkamp,

We are pleased to inform you that your manuscript has been judged scientifically suitable for publication and will be formally accepted for publication once it complies with all outstanding technical requirements.

With kind regards,

Andre van Wijnen

Academic Editor

PLOS ONE

Additional Editor Comments (optional):

Reviewers' comments:

Reviewer's Responses to Questions

**Comments to the Author**

1. If the authors have adequately addressed your comments raised in a previous round of review and you feel that this manuscript is now acceptable for publication, you may indicate that here to bypass the “Comments to the Author” section, enter your conflict of interest statement in the “Confidential to Editor” section, and submit your "Accept" recommendation.

Reviewer #2: All comments have been addressed

2. Is the manuscript technically sound, and do the data support the conclusions?

Reviewer #2: Yes

3. Has the statistical analysis been performed appropriately and rigorously? 

Reviewer #2: I Don't Know

4. Have the authors made all data underlying the findings in their manuscript fully available?

Reviewer #2: Yes

5. Is the manuscript presented in an intelligible fashion and written in standard English?

Reviewer #2: No

6. Review Comments to the Author

Reviewer #2: (No Response)

7. PLOS authors have the option to publish the peer review history of their article (what does this mean?). If published, this will include your full peer review and any attached files.

Reviewer #2: No

---

## [Editor Report · Acceptance letter]

19 Aug 2019

PONE-D-19-14765R1 

Atoh8 acts as a regulator of chondrocyte proliferation and differentiation in endochondral bones 

Dear Dr. Vortkamp:

I am pleased to inform you that your manuscript has been deemed suitable for publication in PLOS ONE. Congratulations! Your manuscript is now with our production department. 

With kind regards,

on behalf of

Dr. Andre van Wijnen 

Academic Editor

PLOS ONE